# Arsenic in Hair as a Marker of Exposure to Smoke from the Burning of Treated Wood in Domestic Wood Burners

**DOI:** 10.3390/ijerph17113944

**Published:** 2020-06-02

**Authors:** Kim N. Dirks, Alana Chester, Jennifer A. Salmond, Nicholas Talbot, Simon Thornley, Perry Davy

**Affiliations:** 1Department of Civil and Environmental Engineering, Faculty of Engineering, The University of Auckland, Private Bag, Auckland 92091, New Zealand; 2Faculty of Medical and Health Sciences, The University of Auckland, Private Bag, Auckland 92091, New Zealand; chestera28@gmail.com (A.C.); s.thornley@auckland.ac.nz (S.T.); 3Faculty of Science, The University of Auckland, Private Bag, Auckland 92091, New Zealand; j.salmond@auckland.ac.nz; 4Auckland Council, Private Bag 92300, Victoria Street West, Auckland 1142, New Zealand; niktalbot@gmail.com; 5Institute of Geological and Nuclear Sciences, P.O. Box 30-368, Lower Hutt 5040, New Zealand; p.davy@gns.cri.nz

**Keywords:** arsenic, wood-burning, exposure, heating, house, hair, biomarker

## Abstract

Timber treated with the anti-fungal chemical copper chrome arsenate is used extensively in the New Zealand building industry. While illegal, the burning of treated timber is commonplace in New Zealand and presents a health risk. Outdoor ambient monitoring of arsenic in airborne particulate matter in New Zealand has identified levels that exceed the maximum standards of 5.5 ng m^−3^ (annual average) at some urban locations. In this study, two-week-old beard hair samples were collected during the winter months to establish individual exposure to arsenic using Inductively Coupled Plasma-Mass Spectrometry. These results were then compared with questionnaire data about wood burner use for the two weeks prior to sampling, and spatial trends in arsenic from ambient monitoring. Results suggest that the burning of construction timber that may contain arsenic is associated with a higher level of arsenic in hair than those who burn logs or coal exclusively. There is no association between the area-level density of wood burners and arsenic levels but a significant correlation with individual household choice of fuel as well as the smell of wood smoke in the community, suggesting very localised influences. Strategies are needed to raise awareness of the risks of burning treated timber and to provide economically-viable alternatives.

## 1. Introduction

New Zealand’s damp climate has led to the extensive use of the anti-fungal chemical copper chrome arsenate (CCA) in construction wood. As a consequence, New Zealand is one of the highest consumers of CCA in the world [1]. Ingestion of arsenic has been associated with increased incidents of liver, bladder, and lung cancers, and has been found to decreases the production of red and white blood cells, which may cause fatigue, abnormal heart rhythm, and blood-vessel damage resulting in bruising. Those exposed also experience skin effects such as the growth of sores and warts [2]. When burnt, a large percentage of the arsenic becomes volatilised, allowing it to enter rapidly into the human body, primarily through respiration [3]. Exposure to arsenic through inhalation is associated with sore throats and irritated lungs, as well as an increased risk of lung cancer from the production of free radicals creating cell damage [4]. It has also been reported that high concentrations can also alter DNA repair [5]. As a result, in most situations, the burning of treated wood is illegal in New Zealand, as stipulated in regional plans from around New Zealand, including in Auckland [6]. However, treated timber offcuts from construction or demolition activities are usually freely available and therefore continue to be used as solid fuel for domestic space heating appliances (wood burners), irrespective of legislation. A recent survey carried out in the Auckland suburb of Henderson indicated that 17% of residents with wood burners admitted to using (often treated) timber offcuts as fuel when available [7]. Health-based ambient air quality guidelines (AAQG) for arsenic have been set in New Zealand by the Ministry for the Environment [8] and regionally by Auckland Council under the Auckland Unitary Plan [9]. For arsenic, the AAQG is 0.0055 µg/m^3^ (5.5 ng/m^3^) as an annual average. There are no short-term regulatory guidelines for arsenic exposure. The ambient guideline value for inorganic arsenic is based on an acceptable risk value of 1 in 100,000 for a high-potency carcinogen such as arsenic [8].

Multi-elemental analysis (including arsenic) of filter-based air particulate matter samples have been collected in New Zealand since 1998 to understand and quantify source contributions to particulate matter concentrations [10,11,12]. In most locations monitored, compared with the summer months (December, January, February), higher levels of arsenic were observed in the winter months (June, July, August), when most of the wood-burning occurs, [13,14,15,16,17,18,19,20]. Moreover, arsenic levels have been found to be associated with the chemical composition profile of wood combustion, linking its presence in air specifically to wood burning. For some of the sites monitored, the recorded levels of arsenic exceeded the maximum allowable annual average levels according to the New Zealand Ambient Air Quality guidelines of 5.5 ng/m^3^ [8]. This suggests that arsenic is present at levels of concern for human health in at least some urban areas across New Zealand, including in Auckland [8], and that the burning of treated wood is specifically linked to this exposure pathway.

A limitation of many health studies of environmental toxins is establishing a link between the spatial concentrations of a pollutant in the outdoor environment and the exposure of individuals who live in the community and spend most of their time indoors. In relation to wood smoke, exposure is further complicated by the fact that some of the exposure (most, in some cases) occurs within the homes themselves. Therefore, monitoring of exposure at the individual level is required in order to assess the actual exposure of residents to arsenic and quantify the risk.

As with many heavy metals, arsenic, once inhaled, is deposited in bodily tissues and organs. When a person is exposed to arsenic, the body will remove some of it through excretory pathways. Arsenic that is circulating in the blood will form a covalent complex with sulphydryl groups (SH) of the amino acid cysteine.

The main structural protein in hair and nails, α-Keratin, contains many SH groups and is therefore a major site for the accumulation of arsenic [21]. Once arsenic is incorporated into hair, it is no longer biologically available. Due to this accumulation, concentrations in the hair and nails tend to be higher than in the rest of the body [22,23], thus the sampling of such tissues can be used as a ‘proxy’ for an individual’s dose in response to their exposure to the toxin.

The concentration of arsenic at the root of the hair will be in equilibrium with the concentration in the blood. Because of this, a single hair can give an exposure timeline and indicate when a subject was exposed to higher or lower concentrations depending on the location along the hair’s length and the growth rate. This segmental hair analysis (i.e., the determination of the concentration along the length of the hair) could provide useful information on the timing of acute arsenic exposure [24,25]. A limitation is that hair is often contaminated with hair products and will consist of growth that has occurred over a period of a few months or more depending on its length. In contrast, beard hair collected from someone who is a ‘regular shaver’, and will have been grown over a much more limited timeframe (say a week or so) depending on the shaving frequency. This means that beard hair samples may be matched with the exposure period with a degree of accuracy that cannot be achieved easily with a normal (rest of the head) hair sample. Such very short beard hair is also much less likely to be contaminated with hair products.

The aim of this study is to help fill a gap in knowledge about people’s exposure to arsenic from direct inhalation through individual household wood burner behaviour and the relative arsenic dose of those living in the households. The study also aims to determine whether there was a discernible spatial variability in arsenic levels across the city that could be linked to wood-burning behaviour at an area level. Personal exposure to arsenic is measured by the chemical analysis of beard hair for arsenic concentration, information on wood burning behaviour is provided by the administration of a short questionnaire to the same participants who provided the hair, and arsenic concentrations in the outdoor air provided through sampling from fixed ambient air quality monitoring stations.

## 2. Materials and Methods

### 2.1. Participants and Sampling

Study participants were recruited from Auckland, the largest city in New Zealand, with an area of 1086 km^2^ and a population of about 1.6 million. Participants, all male, were recruited from campuses of the University of Auckland located in central Auckland, and all either lived in the city or lived in Auckland’s suburbs and commuted into the city for study or work. Participants were asked to shave their beard completely, grow their beard for two weeks, and then collect the resulting growth. Beard hair samples were collected by the individual volunteers, as per the instructions provided in a collection kit and using electric clippers and/or a razor. Participants were instructed to collect only hair that was clean, dry and free from oils/shaving cream. Participants were also asked to complete a brief questionnaire providing information about their age, smoking activity (including flatmates, partners, etc.), their tap water consumption (a potential source of arsenic), their suburb, their wood burner usage, the fuel type (if used), and the frequency of wood smoke presence in their neighbourhood as perceived by smell. Participants also included the date of the start of the beard growth, as well as the date of collection. The hair sample (sealed in a paper envelope) and the completed questionnaire were returned at the same time so the results could later be matched. Ethical approval for the study was obtained from the University of Auckland Human Participants Ethics Committee on 12 June 2018 (Number 021366).

### 2.2. Chemical Analysis of Hair Samples

Measurements of concentration of generic/total arsenic (in µg per g hair) were made in-house (using university facilities) on an Inductively Coupled Plasma-Mass Spectrometry instrument (ICP-MS) in Helium mode to reduce polyatomic interferences. Calibration standards were prepared in a matrix-matched solution from 1000 ppm single element standards (Peak Performance, CPI International).

### 2.3. Analytical Methods

Prior to testing the beard samples, a trial was run using donated hair from one individual to test a number of washing methods as current literature does not indicate a standard method [26]. This test was to determine which method resulted in the least amount of lost analyte. Exposure to heavy metals through the air can result in both endogenous (metals metabolically incorporated into hair structure during its growth process) and exogenous (ambient metals adhering to hair surface) traces in hair [26]. The samples were not washed prior to analysis in order to reflect full contaminant exposure due to pollutants in ambient air.

The sample was weighed (~100–300 mg) into a 100 mL Teflon tube, and 2 mL of 69% Tracepur HNO_3_ (Merck, Auckland, New Zealand) and 2 mL of 35% H_2_O_2_ (ECP Ltd., Auckland, New Zealand) were added. The vessels were sealed and placed in a Maxi-44 rotor and digested in an Ethos-Up Microwave reaction system (Milestone SRL, Sorisole, Italy) at 180 °C for 20 min. This involved 20 min of a ramp in temperature from ambient to 180 °C and a cooling period back to ambient temperature over a period of approximately 35 min. The digest was diluted with 21 mL of Type-1 deionized water and mixed, and a final solution weight was obtained and divided by the sample mass to calculate the total dilution factor. While three of the participants provided below the ideal minimum amount of hair of 100 mg required, the analysis was nonetheless carried out but the sample numbers noted. One participant did not provide sufficient hair; thus their sample was not included in the analysis.

### 2.4. Quality Assurance and Control

The reagent blank sample was digested in the same way as the hair samples, and the value was used to correct the instrument readings. The method’s limit of detection (MLOD) was calculated by the instrument during calibration. The MLOD for arsenic was 9.29 × 10^−6^ µg/g. There is no certified reference material for unwashed hair [26], so none was used. Instead, an online internal standard (20 ppb Tb) was used to monitor and correct for instrument drift and matrix effects. The methodology used for this study has precedent [26], and was deemed best-suited to a research scope that did not target exact As concentrations, rather, assessed and reported on whether beard samples could be used as a biomarker for As and correlated to the burning of CCA treated timber in urban dwellings. The recoveries of this standard for each sample ranged from 94% to 105%.

### 2.5. Area-Level Wood Burner Density

Figure 1 shows the spatial distribution of wood burners that use solid fuel in the Auckland Region. It indicates that the density is highly spatially variable but tends to be higher in the outer suburbs near the Auckland Region’s border (indicated by the black line). For the neighbourhoods reporting a high density of wood burners (from 2.41–3.27 households per hectare), there is an increased risk of exposure to arsenic if any of the houses in the neighbourhood burn treated timber. For this reason, the levels of arsenic found in the beard hair samples were compared with the density of wood burners reported upon in the New Zealand 2013 Census [27] at the census area unit level. This is based on the information provided in the completed questionnaire about the suburb in which participants lived.

### 2.6. Statistical Analysis

Data analysis was performed using R (version 3.6.3, The R Foundation for Statistical Computing, Vienna, Austria) [28]. The analysis consisted of the analysis of variance to test for differences between types of fuel used and concentrations of arsenic, and Fisher’s exact test to determine the extent of usage of the wood burner, the smell of smoke, smoking status and categories of wood burner densities on the type of wood burner to determine whether these could explain the differences in arsenic concentrations. Alpha (false-positive) levels were set at 5% for all statistical tests.

## 3. Results

The study sample consisted of 31 participants. Two smokers, two that lived with smokers, and 27 non-smokers. One of the smokers reported being a very heavy smoker and was considered an outlier and not included in the subsequent analyses. Participants varied in age from 20 to 65 years. Across all of the measurements, the mean (±standard deviation) concentration was found to be (0.071 ± 0.053) µg/g, with a maximum value of 0.23 µg/g and a minimum value of 0.017 µg/g. Arsenic concentration was right-skewed and so transformed for regression using a natural logarithm. Both the age of subjects and their reported water intake were not significantly associated with arsenic concentration. Univariate analysis of the association between age (years) and transformed arsenic concentration yielded an exponentiated beta coefficient of 1.03 (95% CI: 0.99 to 1.07), meaning that each year of age was associated with a 3% increase in arsenic concentration (*p* = 0.208). Similarly, water intake (transformed to the midpoint of categories), was not significantly associated (exponentiated beta coefficient: 1.43; 95% CI: 0.75 to 1.43).

Table 1 shows the study results of arsenic concentration in relation to the factors considered in the study, including wood burner usage, the presence of a smell of woodsmoke, smoking status of the study participant and household, as well as the density of wood burners in the participant’s neighbourhood, based on data collected in the participant questionnaire. Arsenic concentration was strongly associated with the fuel used, being highest in the ‘mix’ fuel group. Of the 30 participants, 11 had wood burners that were used during the sampling (beard growth) period. Of these 11, one used coal, six used exclusively untreated logs, three used a mix of logs and offcuts, and one used only offcuts as fuel. For the purpose of classification, the participant who burned coal was considered to burn exclusively logs (i.e., no possibility of arsenic content), and the participant who burned exclusively offcuts was considered to burn a mix of fuels (i.e., timber with the potential for arsenic content). Of the 30 participants, 10 reported never smelling wood smoke in their neighbourhood, 12 reported sometimes smelling wood smoke and eight reported that smelling wood smoke was common. Geographically, the subjects were well distributed across Auckland in terms of the suburbs in which their residences were located. Of the participants, 5, 18 and 7 lived in areas with a wood burner density of 0–1, 1–2 and 2–3 per hectare, respectively.

As seen in the tables above, arsenic concentrations vary significantly with wood burner usage (or not), the frequency of wood smoke observed, and the type of fuel used. Figure 2 shows arsenic concentrations with map and boxplots sorted by wood burner usage, and wood smoke frequency. Figure 3 shows a boxplot for the mean concentration by fuel type. From this small sample size, we cannot draw any major conclusions, but the most prevalent factor influencing arsenic concentrations in this group was the frequency of wood smoke observed. With a *p*-value of 0.009 and an eta squared of 29.7%, this suggests that almost 30% of the variability in the concentrations could be due to the prevalence of wood smoke in the subject’s immediate neighbourhood. Pairwise comparisons within the group were used to show that not only is there a significant difference between those who did not smell smoke and those who did (*p* = 0.026), there was also a significant difference between those who smelled smoke sometimes and those who reported the smell of woodsmoke frequently (*p* = 0.003). Note that there was no significant difference found between the area-level density of wood burners (from the Census data of Figure 1) and the levels of arsenic in the beard hair of participants (see Figure 4). This suggests that the associations are very localised.

The type of fuel used was also a significant factor with *p* = 0.016 and eta squared 26.4%. However, using pairwise comparisons within the group, this ‘significant’ difference is due to the difference between the ‘Mix’ subjects and the ‘None’ subjects (*p* = 0.006). There was no significant difference between the ‘Mix’ and the ‘Logs’ groups (*p* = 0.161), which indicates that those burning offcuts, or a mix of logs and offcuts resulted in significantly higher arsenic levels than those burning nothing at all, but not significantly higher than those burning just logs.

Finally, the presence or absence of a wood burner in the home had a smaller effect on the personal exposure with *p* = 0.046 and eta squared of 13.5%. These results indicate that people using wood burners are exposed to higher concentrations of arsenic, especially if they choose to burn treated wood. However, more revealing is that community behaviour illustrated by wood smoke in the immediate neighbourhood was the more significant factor in arsenic exposure for this sample population, and not the density of wood burners at the census (wider) area level.

## 4. Discussion

At present, there are very few published results on the identification of arsenic inhalation specifically through the burning of treated timber. This means that the results reported here cannot be directly compared to others. However, the use of scalp hair samples to test for ingestion of toxic elements has been previously reported and the results of the studies are summarised in Table 2. The methodology of pre and post washing of submitted beard samples in this study removes some of the uncertainties described in previous research [29] where reported external contamination of arsenic could not be distinguished from ingested concentrations within the hair strands. The limitation in quantifying exposure to arsenic using hair as an indicator of absorbed dose is the fact that the concentration of arsenic in the hair over the head of the same person can vary significantly [30]. In the current study, hair from individual subjects was analysed by testing different washing methods- and the duplicates that were produced did not always agree. It would hold that beard hair is similar in this trait though this remains unverified. As such, if we obtain low concentrations in a sample, it does not necessarily mean that the subject was not exposed to higher concentrations, only that this portion of hair did not accumulate to a high concentration. Conversely, if we obtain high concentrations, it would be indicative that the subject was in fact exposed to higher concentrations. Through the use of beard samples for this study, this concern can be minimised with fresh growth likely to occur in just a matter of few days, leading to a small amount of variability in the sample.

Although the sample size for this study was too small to extract meaningful social data, the distance from central Auckland appears to be a factor, with higher concentrations found further away from the city centre. This could be due, at least in part, to the higher density of wood burner usage in the areas further away from the city centre [31]. Moreover, home-ownership and household wealth have also been found to be contributing factors to home heating choice [6], though this was not investigated further here.

Many other studies report underlying demographic trends within those exposed to environmental arsenic. For example, scalp-grown hair for environmental toxins has previously been reported with a mean of 1.53 µg/g [32]. This result was anomalously high when compared to those found in the present study and elsewhere. It may be because of a disproportionate number of smokers having to be part of the sampled group. An anomalously high concentration was recorded during the present study (24.92 µg/g), a value associated with the participant who indicated he was a heavy smoker. The study by Kumakli et al. [32] also assessed the influence of gender, smoking habits, and age on the elemental concentrations in scalp hair. Higher concentrations of essential elements were observed in the scalp hair of females and non-smokers, and the differences were often significant at a 90% confidence level. This study was limited to male participants so no gender analysis was possible.

Research carried out in Sicily [36], using a similar scalp hair technique as used previously [32], identified raised arsenic levels in children living close to the volcanic areas, those living next to quarries and those who reside near a well-known industrial area. Extracting socioeconomic statistics from the ingestion of arsenic from the burning of treated timber is important to local councils. Survey reports carried out by Auckland Council show that poorer demographics and certain ethnicities are more likely to burn treated timber within the Auckland region [7] and would enable policy-makers to focus educational awareness programs in communities where the highest exposure concentrations are recorded. Given the small sample size and the opportunistic sampling methodology (university students recruited through flyers posted on the university campus), it is not possible to generalise the findings to the population as a whole. Nonetheless, the findings do suggest that those living in areas where treated timber is burned may be more at risk.

Given the lack of correlation between the census area average wood burner densities and concentrations and the strong link with both the smell and personal behaviour of the participant, the result suggests that the effects are very localised. This is supported by a prior study in New Zealand that showed down-valley concentration gradients of wood (and treated timber) combustion-related pollutants (PM_2.5_, Black carbon, As, etc.) due the combination of katabatic drainage and cumulative emissions as air mass flows passed across emissions from residential dwellings using wood burners [16,18,37].

A limitation of the study was the reliance on questionnaire data to identify whether wood containing arsenic had been burned. Ethical considerations around illegal behaviour restricted questioning on whether there was the burning of construction wood offcuts, irrespective of whether it had been treated (and therefore whether or not an illegal activity had occurred). Furthermore, work with focus groups in a wood-burning community [38] suggests treated but aged wood or treated wood *per se* is not always easy to identify so individuals may be burning treated timber unknowingly. Analysis of ash collected from participant’s fireplaces, as well as or instead of a simple questionnaire would have indicated more reliably whether or not the wood burned had contained arsenic.

While beard hair remains largely untested with respect to its usefulness as a biomarker for heavy metal exposure, it none-the-less shows promise given the significant effects observed in a very small sample. There are a number of other limitations associated with the study, particularly the small (opportunistic) sample which limits our ability to extrapolate to the general population in any meaningful way. A much larger study with random sampling would be required for this. The fact that beard hair can only be obtained from adult men also has obvious limitations in terms of drawing conclusions about the population as a whole. In such a sample, bias may be introduced due to the gendered aspect of occupational exposures and the handling of wood.

## 5. Conclusions

Increased arsenic concentrations were observed in the beard hair of those burning wood for home heating. Construction timber offcuts, sometimes used in the mix, often contain CCA treated timber. The sampling of beard growth rather than scalp-grown hair allowed for more recent exposures to be sampled, resulting in a more accurate measure of the mean of each sample per subject. Furthermore, the ease with which people can clean beards when compared to longer scalp-grown hair, for example, reduced the risk of external contamination from arsenic rather than through ingestion. However, collecting beard samples limits the diversity of the sample, with potentially important demographic dimensions unable to be quantified. Taking the limited size of the sample into account, the results obtained in this study indicate that this methodology is effective in observing the ingestion of hazardous arsenic by inhalation for those that burn treated timber within the home. The small sample size obviously invites that caution is applied when extrapolating results and drawing conclusions. Further work is required to investigate the local conditions, mobility and behaviour patterns of those with the highest personal exposure, and to place these factors against local emission sources and sociodemographic considerations such as deprivation indices. Activities to improve the understanding of the risks associated with exposure to arsenic to dissuade people from burning wood offcuts based on environmental sustainability and health (warm house) considerations would be worthwhile.

## Figures and Tables

**Figure 1 ijerph-17-03944-f001:**
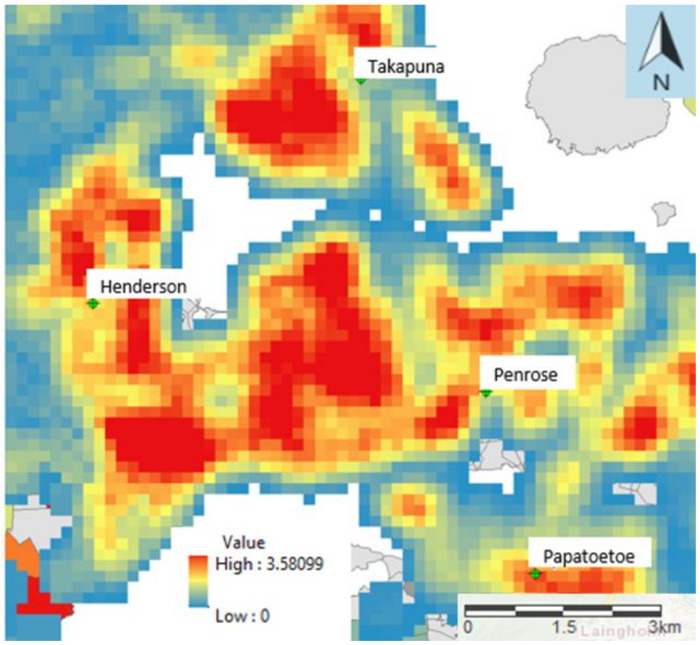
Map of average wood burner per hectare for each census area unit across the Auckland Region. The image has been created using the kernel Density function in ArcGIS. The Kernel density uses kernel function within GIS to fit a smoothly tapered surface to each input of burner density per Census Unit Area (Census 2013).

**Figure 2 ijerph-17-03944-f002:**
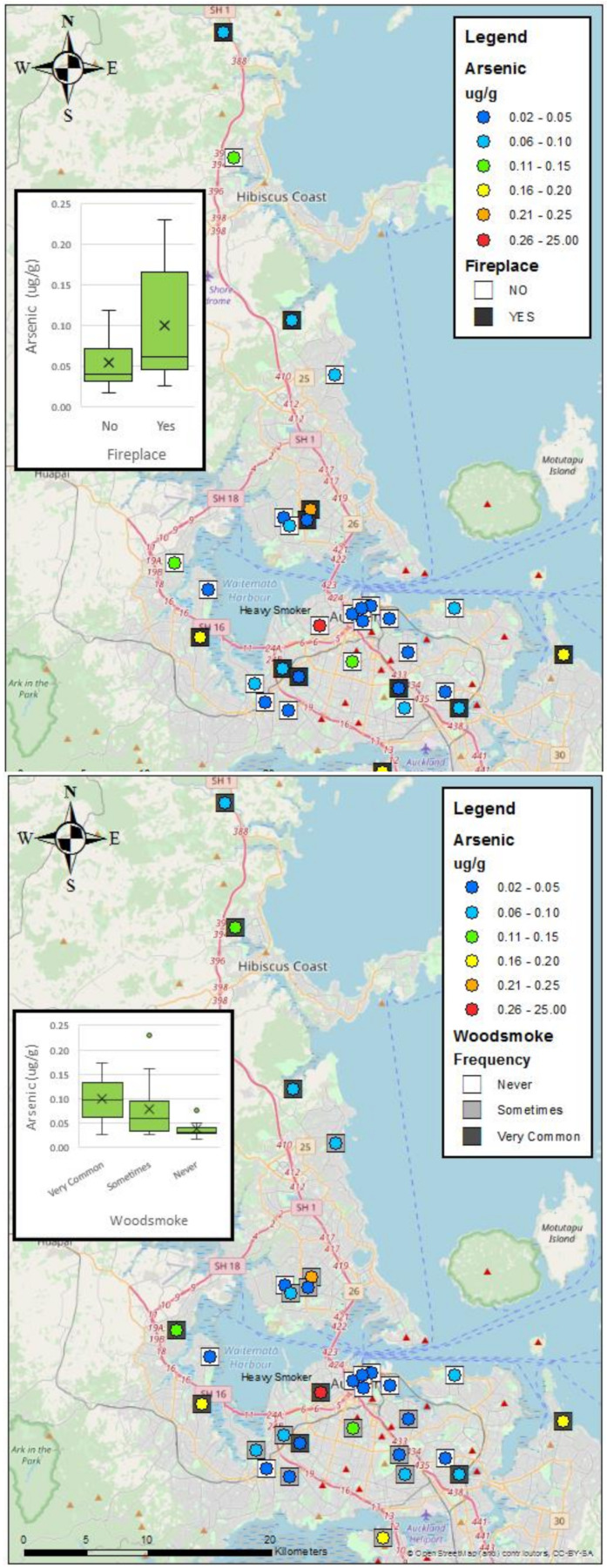
The map of study participants shows the estimated location, as well as the concentration of arsenic found in the beard sample. The left panel includes data on fireplace usage and the right panel shows the frequency of wood smoke present in neighbourhoods based on smell.

**Figure 3 ijerph-17-03944-f003:**
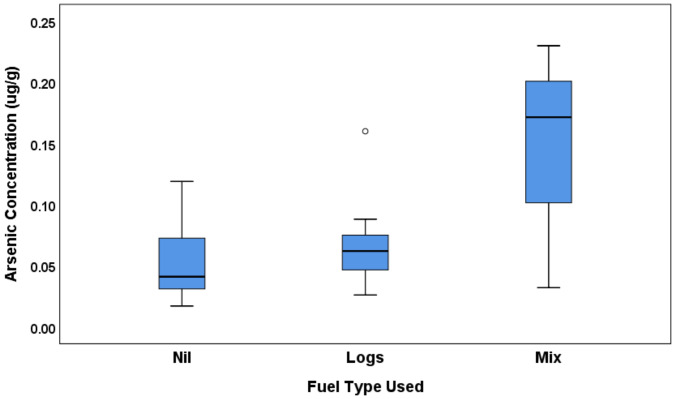
Average arsenic concentrations as a function of the fuel type used.

**Figure 4 ijerph-17-03944-f004:**
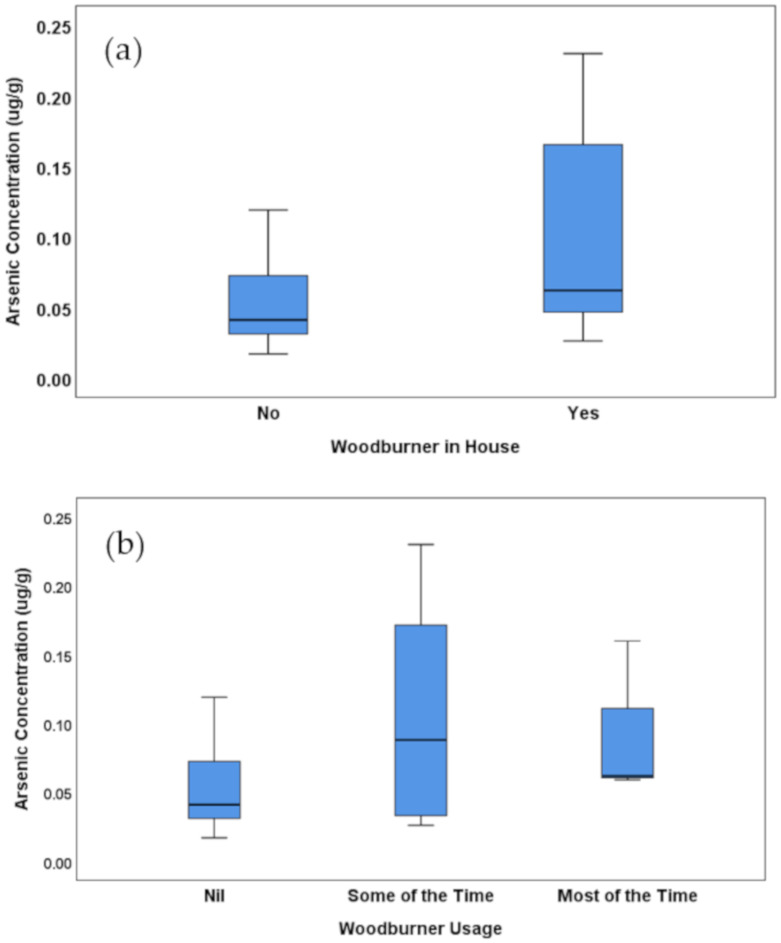
Average arsenic concentrations as a function of (**a**) whether or not the participant has a wood burner in their home, (**b**) the extent of usage of the wood burner (if they have one), (**c**) the extent of woodsmoke smell in their neighbourhood and (**d**) the density of wood burners in the neighbourhood (based on Census 2013 data).

**Table 1 ijerph-17-03944-t001:** Arsenic concentrations of beard hair and other characteristics of subjects by fuel type used.

Factor	None (Column %) *	Logs (Column %) *	Mix (Column %) *	Total (Column %) *	Test Stat.	*p*-Value
**Total**	20	6	4	30		
**Arsenic Concentration**					ANOVA F-test	0.001
**Mean (µg /g) (SD)**	0.053 (0.031)	0.078 (0.044)	0.151 (0.084)	0.071 (0.053)		
**Usage**					Fisher’s exact test	<0.001
**None**	19 (95.0)	0 (0.0)	0 (0.0)	19 (63.3)		
**Some**	1 (5.0)	3 (42.9)	4 (100)	7 (23.3)		
**Most**	0 (0.0)	4 (66.7)	0 (0.0)	4 (13.3)		
**Smell of Smoke**					Fisher’s exact test	0.010
**Never**	10 (50.0)	0 (0.0)	0 (0)	10 (33.3)		
**Sometimes**	7 (35.0)	3 (42.9)	2 (50.0)	12 (40.0)		
**Common**	2 (10.5)	4 (57.1)	2 (50.0)	8 (26.7)		
**Smoker**					Fisher’s exact test	0.565
**No**	17 (89.5)	7 (100)	3 (75.0)	27 (90.0)		
**Household**	1 (5.3)	0 (0.0)	1 (25.0)	2 (6.7)		
**Yes**	1 (5.3)	0 (0.0)	0 (0.0)	1 (3.3)		
**Wood Burner Density**					Fisher’s exact test	0.574
**0–1**	3 (15.8)	2 (28.6)	0 (0.0)	5 (16.7)		
**1–2**	11 (57.9)	3 (42.8)	4 (100)	18 (60.0)		
**2–3**	5 (26.3)	2 (28.6)	0 (0.0)	7 (23.3)		

* Unless otherwise indicated. SD: standard deviation.

**Table 2 ijerph-17-03944-t002:** Published average concentrations of arsenic.

Location	Arsenic Concentration (µg g^−1^)	Reference
USA	1.53	Kumakli et al. (2017) [32]
China	0.18 ± 0.17	Luo et al. (2014) [33]
Tunisia	0.02 ± 0.02	Nouioui et al. (2018) [34]
China	0.127 ± 0.078	Liang et al. (2017) [35]
New Zealand	0.071 ± 0.053	Current study

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
