# Peer review of "Arsenic in Hair as a Marker of Exposure to Smoke from the Burning of Treated Wood in Domestic Wood Burners"

_ijerph, 2020, doi:10.3390/ijerph17113944_

Round 1
Reviewer 1 Report
The study tries to understand how the arsenic beard concentration changes among New Zealand subjects for which it seems that the main route of arsenic exposure is the inhalation of burnt wood.
The study is certainly interesting at the local level. However, it has weaknesses, such as the lack of a comparison between genders as the sample is made up exclusively of men. In addition, the sample is limited, only about 30 subjects.
The text should be revised and corrected in the form as not quite specific terms are used.
Overall the work is fine but in the introduction I would try to expand the part related to the arsenic toxicity in general, what happens when it is inhaled and what implications this can have on human health. Otherwise the motivation that leads to assess the presence of arsenic in the beard it is not clear. I think the study was firstly carried out to understand how the arsenic beard concentration varies and then in the future to evaluate if this exposure can give adverse effects on human health.
Major revision
It is requested to expand the introduction regarding the arsenic toxicity, what happens to arsenic once inhaled and what implications it can have on human health.
Please, specify in the materials and methods the form of the speciated arsenic (in this case it is generic arsenic) and the measure unit.
It is requested to better explain that the WHO establishes the threshold limits of inhaled arsenic but then the nations have different limits depending on the type of exposure. Maybe you could try to go deeper into the New Zealand situation.
Also report some results on the arsenic concentration in the samples (min, max, average ...).
Check the tables; they should be self-reading.
Minor revisions
Abstract
- Line 19 - Please delete “200 Max”;
- Line 21 – It is the first time the word “arsenic” is used; please, add and define the acronym here (“As”) and delete the other entire words “arsenic” throughout the text;
- Line 25 – Please delete the acronym “ICP-MS” since it is used just ione time.
Introduction
- Line 39 – The word “chemical” is referred to CCA? If so, please replace the word “chemical” with CCA;
- Line 39 - It is the first time the word “arsenic” is used; please, add and define the acronym here (“As”) and delete the other entire words “arsenic” throughout the text;
- Line 40-41 – “…allowing for easy and rapid ingestion into the human body, primarily through 40 respiration”; “ingestion” means through the mouth…please change the phrase, maybe it could be “allowing it to rapidly enter the human body, primarly…..;
- Line 51 – Please replace “undertaken” with “collected”;
- Line 52-53 – “…, the samples collected have indicated higher levels of arsenic…”; please rewrite the sentence as indicated “In most locations monitored, compared with the summer months (December, January, February), higher levels of arsenic in the winter months (June, July, August), when wood burning mostly occurs, were observed”.
- Line 58 – please specify the acronym “NZAAQ”;
Material and methods
- It is not clear what arsenical form has been speciated through ICP-MS. Inorganic As? Trivalent/pentavalent? Or it has been identified just arsenic?
Results
- Please, at the beginning of the paragraph start providing the total sample number and then it could be specified if they are smoker or not.
- there is no characterization of the arsenic values found in the beard, average minimum maximum value;
- Table should be self-reading; in table 1, what does it mean “Nil”? It means “No use?”, And “col. %”? Please specify better. The measurement unit of the arsenic concentration is not specified.
- Line 184 – Please replace “tables” with “table”.
Author Response
Reviewer 1
The study tries to understand how the arsenic beard concentration changes among New Zealand subjects for which it seems that the main route of arsenic exposure is the inhalation of burnt wood.
The study is certainly interesting at the local level. However, it has weaknesses, such as the lack of a comparison between genders as the sample is made up exclusively of men. In addition, the sample is limited, only about 30 subjects.
The text should be revised and corrected in the form as not quite specific terms are used.
Overall the work is fine but in the introduction I would try to expand the part related to the arsenic toxicity in general, what happens when it is inhaled and what implications this can have on human health. Otherwise the motivation that leads to assess the presence of arsenic in the beard it is not clear. I think the study was firstly carried out to understand how the arsenic beard concentration varies and then in the future to evaluate if this exposure can give adverse effects on human health.
Major revision
It is requested to expand the introduction regarding the arsenic toxicity, what happens to arsenic once inhaled and what implications it can have on human health.
Thank you for your suggestion. We have added a few sentences to flesh out the impacts on health resulting from arsenic exposure, mentioning ingestion and contact as well as inhalation (Lines 35-42):
“Ingestion of arsenic has been associated with increased incidents of liver, bladder, and lung cancers, and has been found to decreases the production of red and white blood cells, which may cause fatigue, abnormal heart rhythm, blood-vessel damage resulting in bruising. Those exposed also experience skin effects such as the growth of sores and warts [2]. When burnt, a large percentage of the arsenic becomes volatilised, allowing it to enter rapidly into the human body, primarily through respiration [3]. Exposure to arsenic through inhalation is associated with sore throats and irritated lungs, as well as an increased risk of lung cancer from the production of free radicals creating cell damage [4]. It has also been reported that high concentrations can also alter DNA repair [5]”.
Please, specify in the materials and methods the form of the speciated arsenic (in this case it is generic arsenic) and the measure unit. Specified in the methods section
An addition has been made to Line 112 of the text:
“Measurements of concentration of generic arsenic (in µg per g hair) were made”
It is requested to better explain that the WHO establishes the threshold limits of inhaled arsenic but then the nations have different limits depending on the type of exposure. Maybe you could try to go deeper into the New Zealand situation.
Please see lines 48-52 where we include some relevant local information with regard to air quality:
“Health-based ambient air quality guidelines (AAQG) for arsenic have been set in New Zealand by the Ministry for the Environment [8] and regionally by Auckland Council under the Auckland Unitary Plan [9]. For arsenic, the AAQG is 0.0055 µg/m3 (5.5 ng/m3) as an annual average. There are no short-term regulatory guidelines for arsenic exposure. The ambient guideline value for inorganic arsenic are based on an acceptable risk value of 1 in 100,000 for a high-potency carcinogen such as arsenic [8].”
Also report some results on the arsenic concentration in the samples (min, max, average ...).
The following details have been added to Lines 161-163:
“Across all of the measurements, the mean (± standard deviation) concentration was found to be (0.071 ± 0.053) µg/g, with a maximum value of 0.23 µg/g and a minimum value of 0.017 µg/g”.
Check the tables; they should be self-reading.
Thank you. We have revised Table 1 as follows: “Nil” has been replaced with “None”. The units for arsenic concentration have been added (µg/g), and ‘Col%’ has been changed to ‘Column %’.
Minor revisions
Abstract
Line 19 - Please delete “200 Max”;
Thank you – done
Line 21 – It is the first time the word “arsenic” is used; please, add and define the acronym here (“As”) and delete the other entire words “arsenic” throughout the text;
Agreed, this in inconsistent. All references to As have now been changed back to ‘arsenic’ throughout for consistency.
Line 25 – Please delete the acronym “ICP-MS” since it is used just ione time.
Done.
Introduction
Line 39 – The word “chemical” is referred to CCA? If so, please replace the word “chemical” with CCA;
Done.
Line 39 - It is the first time the word “arsenic” is used; please, add and define the acronym here (“As”) and delete the other entire words “arsenic” throughout the text;
We have changed al references to As to arsenic, for consistency, including in the abstract.
Line 40-41 – “…allowing for easy and rapid ingestion into the human body, primarily through 40 respiration”; “ingestion” means through the mouth…please change the phrase, maybe it could be “allowing it to rapidly enter the human body, primarly…..;
Thank you. This has been changed to “allowing it to enter rapidly into the human body, primarily through respiration”. (Line 39)
Line 51 – Please replace “undertaken” with “collected”;
Done.
Line 52-53 – “…, the samples collected have indicated higher levels of arsenic…”; please rewrite the sentence as indicated “In most locations monitored, compared with the summer months (December, January, February), higher levels of arsenic in the winter months (June, July, August), when wood burning mostly occurs, were observed”.
Thank you. We have changed the lines to (Lines 55-57):
“In most locations monitored, compared with the summer months (December, January, February), higher levels of arsenic were observed in the winter months (June, July, August), when most of the wood burning occurs, [10-17].”
Line 58 – please specify the acronym “NZAAQ”;
This has been changed to “New Zealand Ambient Air Quality Guidelines” (Line 60)
Material and methods
It is not clear what arsenical form has been speciated through ICP-MS. Inorganic As? Trivalent/pentavalent? Or it has been identified just arsenic?
An addition has been made to Line 112 of the text.
“Measurements of concentration of generic arsenic (in ug per g hair) were made”
Results
Please, at the beginning of the paragraph start providing the total sample number and then it could be specified if they are smoker or not.
We have included the number of participants and then followed this with the number of smokers (see Lines 159-160):
“The study sample consisted of 31 participants. Two smokers, two that lived with smokers, and 27 non-smokers. One of the smokers reported being a very heavy smoker”
There is no characterization of the arsenic values found in the beard, average minimum maximum value
The following details have been added to Lines 162-169:
“Across all of the measurements, the mean (± standard deviation) concentration was found to be (0.071 ± 0.053) µg/g, with a maximum value of 0.23 µg/g and a minimum value of 0.017 µg/g. Arsenic concentration was right skewed and so transformed for regression using a natural logarithm. Both age of subjects and their reported water intake were not significantly associated with arsenic concentration. Univariate analysis of the association between age (years) and transformed arsenic concentration yielded an exponentiated beta coefficient of 1.03 (95% CI: 0.99 to 1.07), meaning that each year of age was associated with a 3% increase in arsenic concentration (P = 0.208). Similarly, water intake (transformed to the midpoint of categories), was not significantly associated (exponentiated beta coefficient: 1.43; 95% CI: 0.75 to 1.43).”
Table should be self-reading; in table 1, what does it mean “Nil”? It means “No use?”, And “col. %”? Please specify better. The measurement unit of the arsenic concentration is not specified.
“Nil” has been replaced with “None”. The units for arsenic concentration have been added (µg/g). The label ‘col %’ have been changed to ‘column %’ for clarity.
Line 184 – Please replace “tables” with “table”
Done.
Reviewer 2 Report
This manuscript has interesting points and contains important data .
However, the following recommendations should be solved .
1.It is necessary to consider the epidemiological variables that can affect the concentration of arsenic in hair, For example, performing a univariate analysis of related variables such as age, water comsumption, and stratification analysis on the related varables.
2. If possible, the authors need to provide the information of the external quality control of hair arsenic( for example, G-EQUS).
Author Response
Reviewer 2
This manuscript has interesting points and contains important data.
However, the following recommendations should be solved.
1.It is necessary to consider the epidemiological variables that can affect the concentration of arsenic in hair, for example, performing a univariate analysis of related variables such as age, water consumption, and stratification analysis on the related variables.
Thank you for your suggestion. The following lines have been added to the text (Lines 161-169):
Arsenic concentration was right skewed and so transformed for regression using a natural logarithm. Both age of subjects and their reported water intake were not significantly associated with arsenic concentration. Univariate analysis of the association between age (years) and transformed arsenic concentration yielded an exponentiated beta coefficient of 1.03 (95% CI: 0.99 to 1.07), meaning that each year of age was associated with a 3% increase in arsenic concentration (P = 0.208). Similarly, water intake (transformed to the midpoint of categories), was not significantly associated (exponentiated beta coefficient: 1.43; 95% CI: 0.75 to 1.43).”
- If possible, the authors need to provide the information of the external quality control of hair arsenic (for example, G-EQUS).
I am sorry but we do not know what G-EQUs is. We have done a search if it and nothing was revealed…
Reviewer 3 Report
The paper presented some data on arsenic levels on the beard hair of New Zealand residents in relation to the As treated wood combustion. The paper is relevant to the journal, in my opinion. I recommend the following amendments:
1) Although there is no reference material of unwashed hair as it was mentioned, washing does not affect the matrix of the hair that much to make the normal hair reference materials inapplicable for the current study. It would be still much better than just running the internal standard to see into the MS drifts. This doesn't validate hair digestion at all! If the authors are not confident in BCRs or SRMs for trace elements in hair, they can also spike some As into their real sample and evaluate the recovery.
2) Please indicate the number of participants in section 2.1
3) Isn't small sample size a limitation (Lines 278-286)?
MISC:
Line 40 Ingestion is normally peroral intake, use term exposure or intake instead here
Line 181 Typo, two full stops.
Line 193 Avoid using contractions in academic context (did not).
Author Response
Reviewer 3
The paper presented some data on arsenic levels on the beard hair of New Zealand residents in relation to the As treated wood combustion. The paper is relevant to the journal, in my opinion. I recommend the following amendments:
- Although there is no reference material of unwashed hair as it was mentioned, washing does not affect the matrix of the hair that much to make the normal hair reference materials inapplicable for the current study. It would be still much better than just running the internal standard to see into the MS drifts. This doesn't validate hair digestion at all! If the authors are not confident in BCRs or SRMs for trace elements in hair, they can also spike some As into their real sample and evaluate the recovery.
Thank you for your suggestion. This would appear to be a recommendation methodology for future hair sample studies and has been gratefully noted by authors.
2) Please indicate the number of participants in section 2.1
Thank you. We added the number of participants to line 159, in Results (Section 3) where we include detail about the participants.
3) Isn't small sample size a limitation (Lines 284-287)?
Yes indeed. The small sample size is mentioned as a limitations of the study in Line 288.
”There are a number of other limitations associated with the study, particularly the small (opportunistic) sample which limits our ability to extrapolate to the general population in any meaningful way. A much larger study with random sampling would be required for this.”
MISC:
Line 40 Ingestion is normally persoral intake, use term exposure or intake instead here
Thank you for your suggestion. This line has been changed to
“allowing it to enter rapidly into the human body, primarily through respiration” (Line 39)
Line 181 Typo, two full stops.
I am sorry, I was not able to find this in the text.
Line 193 Avoid using contractions in academic context (did not).
Thank you. This has been changed to ‘did not’.
Round 2
Reviewer 3 Report
Well, I still have concerns on the use of reference materials in the current study so I leave the decision for the editor.
Author Response
Thank you for your comments. Unfortunately, we are not able to do any further laboratory analysis at this stage due to issues with access to labs. It is certainly something we will look into when we move on to our follow up study.